# Screening of a Small Number of Italian COVID-19 Syndrome Survivors by Means of the Fatigue Assessment Scale: Long Covid Prevalence and the Role of Gender

**Antonella Serafini** [1,*]**, Alberto Tagliaferro** [2,3,*]**, Francesco Balbi** [1]**, Alberto Bordo** [1]**, Stefano Bernardi** [1]**, Giorgio Berta** [1]**, Lorenzo Trucco** [1]**, Enrico Perretta** [1]**, Elisa Gualco** [1]**, Patrizia Zoccali** [1] **and Claudio De Michelis** [1]

[1] S.C. Pneumologia—Ospedale Imperia, Via Sant'Agata 57, 18100 Imperia, Italy; f.balbi@asl1.liguria.it (F.B.); a.bordo@asl1.liguria.it (A.B.); s.bernardi@asl1.liguria.it (S.B.); g.berta@asl1.liguria.it (G.B.); loreales@libero.it (L.T.); e.perretta@asl1.liguria.it (E.P.); e.gualco@asl1.liguria.it (E.G.); p.zoccali@asl1.liguria.it (P.Z.); c.demichelis@asl1.liguria.it (C.D.M.)
[2] DISAT, Politecnico di Torino, Corso Duca degli Abruzzi 24, 10129 Torino, Italy
[3] Consorzio Interuniversitario Nazionale per la Scienza e Tecnologia dei Materiali (INSTM), Via G. Giusti 9, 50121 Florence, Italy
* Correspondence: a.serafini@asl1.liguria.it (A.S.); alberto.tagliaferro@polito.it (A.T.)

**Abstract:** Months after healing, a number of COVID-19 syndrome survivors are affected by both psychological and physical limitations. They are mainly troubled with long-term fatigue, which is a crucial aspect of Long COVID syndrome. This paper aims to investigate the level and persistency of fatigue among COVID-19 survivors from the first wave of the pandemic in Western Liguria and to elucidate the role of gender, age, and lifestyle. It also provides data to the scientific community to help drawing a consistent picture of Long COVID syndrome. The patients were requested to fill a Fatigue Assessment Questionnaire twice: (i) a few weeks after discharge from hospital or home confinement and (ii) a few months later. Statistical analysis was carried out on the global scores and on the score for every single item of the questionnaire. The outcome of the two questionnaires were analyzed separately and compared. Females are more affected by fatigue than males. This results holds for both physical and mental fatigue. All the males' fatigue scores were reduced at the second control, while 40% of females worsened it. Home-confined patients showed a higher fatigue score at the first check. In the initial stages of the recovery, patients are more affected by physical, rather than mental, fatigue. This is worth of further investigation as well as the reasons leading to a higher initial fatigue score for home cared patients.

**Keywords:** chronic fatigue; long-COVID; gender issue

## 1. Introduction

A growing number of studies on long term effects of COVID-19 syndrome are becoming available. As it was the case after Severe Acute Respiratory Syndrome (SARS) epidemic [1], a number of COVID-19 patients are affected after discharge by a serious syndrome, characterized by (i) chronic fatigue, (ii) myalgia, (iii) depression, (iv) sleep disturbances, and (v) a plethora of other debilitating symptoms (breathlessness, chest pain, palpitations, sleep disorders . . . ): the "post COVID-19 syndrome" or "Long COVID". 10–20% of patients are not in good health after three weeks from clinical recovery and 1–3% of them are still compromised 12 weeks later [2]. The post-acute care clinical management evidenced a severe disabling physical and mental fatigue among survivors, lasting for a period over six months, in agreement with the characteristics of chronich fatigue (CF) [3]. Fatigue is a "feeling of tiredness, a distaste towards ongoing activity and unwillingness to carry on the activity" or "an unwillingness to continue the task and an increasing fading of focus on it" [4]. Being fatigue a subjective and non-specific physical and psychological dysfunction, it must be considered a symptom, rather than a measurable outcome of the

illness. The COVID related CF clinical picture is rather similar to the "Chronic fatigue syndrome" (CFS), a disease strongly impacting on social routine, working performance and free time habits, with a female to male ratio of 3:1. However, the diagnosis of chronic fatigue (CF) is a challenging one and can be accepted only if other pathologic processes have been ruled out [5]. In the post COVID syndrome the CF affects hospitalized patients as well as patients treated at home with a mild COVID-19 syndrome. CF affects their convalescence path, their ability to recover the basic functional status and eventually their survival [6].

In this paper we report the physical and mental fatigue prevalence among patients affected by COVID-19 first wave (spring 2020) in Imperia area (Western Liguria, North-West of Italy). Patients were admitted to the post-acute outpatient service after the discharge from the COVID Area of the Pneumology Department or after the end of their home confinement. In Italy, as in most other countries worldwide, the first year of the pandemic has been a quite hectic period and the access to rehabilitation procedures have been strongly limited due to the critical epidemiological situation, the particularly widespread nature of the epidemic and the increase in cases throughout the country: the outpatient elective rehabilitation procedures including respiratory ones have undergone a reprogramming and were even suspended for a period. We therefore decided to monitor the fatigue parameter for twelve months to provide a simple indicator of the evolution of convalescence in a highly conditioned historical context. Through such a simple monitoring we were able to screen the patients and address them, if needed, to the specific care treatments, either physical or psychological. Hence, we focused our study on long-term "fatigue", using a standardized questionnaire. The questionnaire was administered at the first visit of the follow up and again after a few more months. In our study, besides the effect of gender and home or hospital care, we considered also the impact of body-mass index (BMI) and of the smoking habit on the severity of Long-COVID Syndrome fatigue. Despite the limited number of patients, our study is able to show as more relevant outcomes that (i) in the initial stages of recovery there is a higher impact on the physical fatigue rather than on the mental one, and (ii) the fatigue affects more heavily females than males at all stages. We also found that females are more keen to develop Chronic Fatigue in Long-COVID Syndrome. These findings, together with others coming from similar studies will contribute to understand and define the path to full recovery of COVID-19 survivors.

## 2. Materials and Methods

We enrolled a set of patients in Imperia area (Western Liguria—North-West of Italy) who were diagnosed with COVID 19 disease between 15 March and 20 April 2020 and discharged from hospital or from home confinement. This study was approved by the relevant Ethical Committee (see Ethical Statement for details) and all patients whose data were considered for this work gave their explicit consent to data utilization. Inclusion criteria: a positive SARS-CoV-2 swab test (RT-PCR test of nasopharyngeal swabs); age $\geq$ 18 years. Exclusion criteria: patients who are unable to read, understand or unwilling to provide the informed consent necessary for participation. The enrolled patients were asked about their physical and mental symptoms fatigue and were requested to fill a questionnaire (Fatigue Assessment Scale (FAS) questionnaire [7]) at every follow up visit, run every 2 to 4 months. For each patient we considered in this study two questionnaires: (i) the first one (filled at the baseline follow up visit), and (ii) the last completed questionnaire at the 12 months deadline. This choice was made in order to compare the evolution of the scores for (i) patients that quickly overcame fatigue (a patient with a score below 22 is considered unaffected by fatigue [8]) and subsequently exited the follow up program and (ii) patients that had more persistent fatigue.

Our sample consisted of 23 patients: 10 females and 13 males, their age ranging from 37 to 76 years (the age values in years were M(59) SD(10) for the whole sample; M(61) SD(10) for males; M(57) SD(9) for females). 14 patients had comorbidities (see Table 1). 13 patients (56.5%) were previous smokers, the remaining 10 (43.5%) were no smokers.

9 patients (39.1%) were treated at home, the remaining 14 (60.9%) have been hospitalized. The home confined patients suffered from mild COVID-19 disease (malaise, fever, dry cough, no respiratory distress). The remaining patients were hospitalized due to respiratory distress, dry cough, dyspnea, failure lung and ground glass and/or consolidative opacities at thorax CT. 7 patients were also treated with oxygen therapy, either conventional or high flow rate, 10 with non invasive mechanical ventilation, 5 of which also with mechanical invasive ventilation because suffering from acute respiratory distress (ARDS) [9].

**Table 1.** Clinical characteristics of the patients.

| Characteristic | Total (%) | Male (%) | Female (%) |
|---|---|---|---|
| Patients | 23 (100) | 13 (56.5) | 10 (43.5) |
| Place of care | | | |
| Hospital | 14 (60.9) | 10 | 4 |
| Home | 9 (39.1) | 3 | 6 |
| Age (years) | M (59) SD (10) | M (61) SD (10) | M (57) SD (9) |
| range | 37–76 | 46–76 | 37–67 |
| Body-mass index | M (28.0) SD (4.9) | M (26.8) SD (3.6) | M (29.5) SD (6.0) |
| No Comorbidities | 9 (39.1) | 4 | 5 |
| Single Comorbidity | 9 (39.1) | 5 | 4 |
| Several Comorbidities | 5 (21.8) | 4 | 1 |
| Comorbidities | | | |
| Arterial hypertension | 5 (21.8) | 2 | 3 |
| Atrial fibrillation | 1 (4.3) | 1 | – |
| Ischemic cardiopathy | 4 (17.4) | 3 | 1 |
| Bronchial Asthma | 2 (8.7) | 2 | – |
| Dislypidemy | 2 (8.7) | 1 | 1 |
| Hipothyroidism | 1 (4.3) | – | 1 |
| OSAS * | 3 (13.0) | 2 | 1 |
| Diabetes mellitus | 1 (4.3) | 1 | – |
| COPD ** | 1 (4.3) | 1 | – |
| Smoking habit | | | |
| Previous Smokers | 13 (56.5) | 7 | 6 |
| No Smokers | 10 (43.5) | 6 | 4 |

* OSAS = Obstructive Sleep Apnea Syndrome. ** COPD = Chronic Obstructive Pulmonary Disease.

The first questionnaire was filled between 30 and 110 days after discharge, the second between 30 and 320 days after the first. The lower intervals between the two questionnaires were usually for patients that recovered (see above), the longer one for patients that did not overcome fatigue (i.e., still had a FAS score $\geq$ 22 after 1 year).

**Fatigue Assessment Scale (FAS)**: Our study is based on the Fatigue Assessment Scale (FAS). Since FAS has been successfully assessed as a valid and reliable measure of fatigue in several diseases [10], we selected FAS as a tool to monitor the impact of fatigue and its evolution in COVID-19 syndrome survivors. The FAS questionnaire consists of 10 questions (Table 2) and is aimed to investigate the presence of fatigue in the Long COVID Syndrome from two different viewpoints: 5 questions investigate Physical Fatigue, the other 5 investigate Mental Fatigue [7]. The answers to the questionnaire are analyzed by attributing numerical values from 1 to 5 to the various answers (see Table 3). Following Michielsen et al. (2003) [7], answers to questions 4 and 10 are reverse scored. The total score, obtained summing up the score of all answers, ranges from 10 to 50. We evaluated the scores also for each of the two Fatigue types: Physical Fatigue (items 1, 2, 3, 4, 5) and Mental Fatigue (items 6, 7, 8, 9, 10). Each score ranges from 5 to 25.

**Table 2.** Fatigue Assesment Syndrome (FAS) questionnaire.

| N | Item |
|---|---|
| 1 | I am bothered by fatigue |
| 2 | I get tired very quickly |
| 3 | I don't do much during the day |
| 4 | I have enough energy for everyday life |
| 5 | Physicall, I feel exhausted |
| 6 | I have problems starting things |
| 7 | I have problems thinking clearly |
| 8 | I feel no desire to do anything |
| 9 | Mentally, I feel exhausted |
| 10 | When I am doing something, I can concentrate very quickly |

**Table 3.** FAS questionnaire scores.

| Answer | Score for Items 1, 2, 3, 5, 6, 7, 8, 9 | Score for Items 4, 10 |
|---|---|---|
| Never | 1 | 5 |
| Sometimes | 2 | 4 |
| Regularly | 3 | 3 |
| Often | 4 | 2 |
| Always | 5 | 1 |

**Statistical analysis**: For each item and for the total FAS scores of the questionnaire we evaluated the average value and its associated standard deviation. As a first step we checked with the Shapiro-Wilk test whether or not the FAS scores were normally distributed. As this is not the case (see below) and the number of patients is limited, we used the Mann-Whitney U test [11] to evaluate the significance of the differences (previous smokers vs no smokers, male vs females, hospitalized vs home cared). To check the relevance of age and BMI on FAS score we evaluated the Pearson's correlation coefficient. The Cronbach's alpha values, evaluated using online resources [12], are 0.90 and 0.97 for the 1st and 2nd questionnaires respectively confirming FAS reliability in the framework we used it.

We have analyzed the outcome considering various factors (BMI, age, gender, smoking habit, place of care). However no effect (see below) was found for BMI, age and smoking habit. Hence, in applying the Bonferroni correction we considered only two factors (gender, place of care). This leads to a threshold significance level $p = 0.025$.

Matt-Whitney U test were carried out using online resources [13]. All power evaluations and Fisher's exact test were carried out using GPower 3.1 software [14]. Online resources were used for Shapiro-Wilk test of normality [15].

## 3. Results

The outcome of the Shapiro-Wilk test show that the FAS scores cannot be assumed to be normally distributed. This is true for both the 1st ($p = 0.09$, due to the high kurtosis: $-1.14$) and the 2nd ($p = 0.002$) questionnaire scores. As we have collected data in two steps, we will discuss at first the outcomes for each step (i.e 1st questionnaire, 2nd questionnaire) and later compare the two questionnaires outcomes to investigate the evolution of fatigue.

### 3.1. 1st Questionnaire

An overview of the total FAS scores is reported in Figure 1 (see also Table S2 in SI). A significant difference exists in FAS score between females and males ($p < 0.001$), with

higher score for females. The presence of a significant gender difference in the FAS score drove us to perform a more detailed analysis of FAS answers per gender, aimed to better elucidate the issue. The comparison between genders of the scores for each answer is reported in Figure 2 (see also Table S3 in SI). Mann-Whitney U test shows that there are significant differences for items 1 ($p = 0.010$), 2 ($p = 0.003$), 4 ($p = 0.006$), 5 ($p = 0.006$), 6 ($p = 0.005$), 7 ($p = 0.005$), 9 ($p = 0.002$) while no significant difference was found for items 3 ($p = 0.132$, $d = 0.56$, power = 0.34), 8 ($p = 0.034$; $d = 1.02$, power = 0.78), and 10 ($p = 0.031$; $d = 0.89$, power = 0.60). In all cases females evidence more fatigue. The outcome (female have significant higher scores at the p < 0.01 level) holds also for the mental ($p = 0.002$) and physical fatigue ($p < 0.001$) subsets.

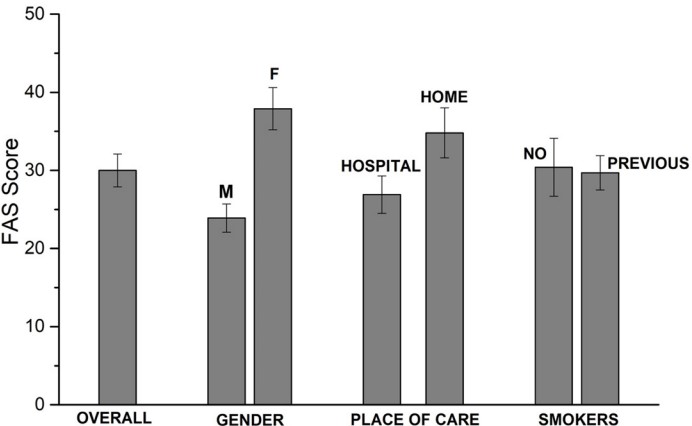

**Figure 1.** 1st questionnaire scores.

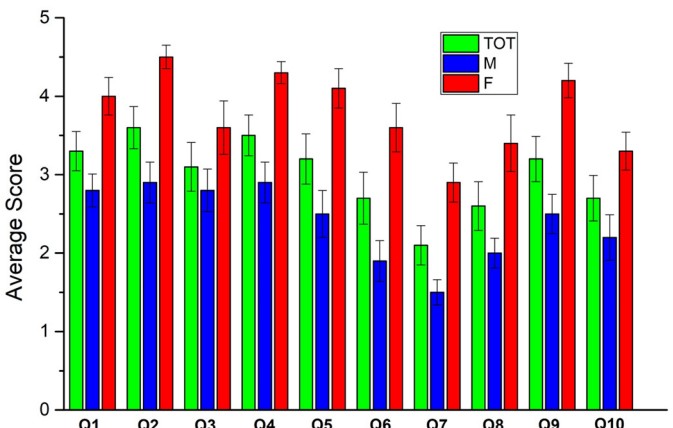

**Figure 2.** 1st questionnaire: average score for each question (total, males, females).

A non significant difference exists between the total scores of hospitalized and home confined patients ($p = 0.0475$, $d = 0.81$, power = 0.56) although home confined ones have nevertheless higher scores. A non significant difference exists for mental ($p = 0.054$, $d = 0.76$, power = 0.52) and physical ($p = 0.034$, $d = 0.77$, power = 0.51) fatigue too. It has to be noted that, given the $p$ values close to the threshold and the large effect size, larger samples will be needed to confirm or deny this result. To remain on the conservative side, we assumed that a false negative cannot be ruled out and included place of care as an impact parameter for Bonferroni correction calculation.

### 3.2. 2nd Questionnaire

An overview of the total FAS scores is reported in Figure 3 (see also Table S5 in SI). A significant difference exists in FAS score between females and males ($p = 0.012$), with higher score for females. The comparison between genders of the scores for each answer is

reported in Figure 4 (see also Table S6 in SI). Mann-Whitney U test shows that there are significant differences for items 1 ($p = 0.005$), 2 ($p = 0.017$), 4 ($p = 0.019$), and 5 ($p = 0.003$) while no significant differences were found for items 3 ($p = 0.154$, $d = 0.68$, power = 0.50), 6 ($p = 0.057$; $d = 0.77$, power = 0.53), 7 ($p = 0.119$; $d = 0.78$, power = 0.59), 8 ($p = 0.026$, $d = 1.01$, power = 0.79), 9 ($p = 0.038$, $d = 1.01$, power = 0.75) and 10 ($p = 0.047$; $d = 0.82$, power = 0.60). For most of them a larger sample will surely provide a more reliable outcome (see comment in the previous paragraph). In all cases females evidence more fatigue. The outcome (female have significant higher scores) holds also for the physical fatigue ($p = 0.007$) while mental fatigue ($p = 0.029$, $d = 1.00$, power = 0.73) does not show a significant difference. No significant difference exists between hospitalized and home confined patients for the total FAS score ($p = 0.206$, $d = 0.47$, power = 0.27), the mental ($p = 0.149$, $d = 0.42$, power = 0.24) and the physical ($p = 0.174$, $d = 0.49$, power = 0.28) fatigue.

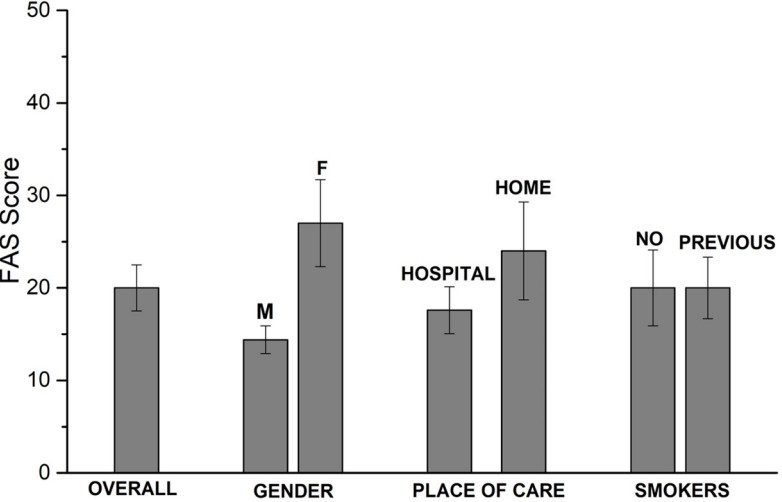

**Figure 3.** 2nd questionnaire scores.

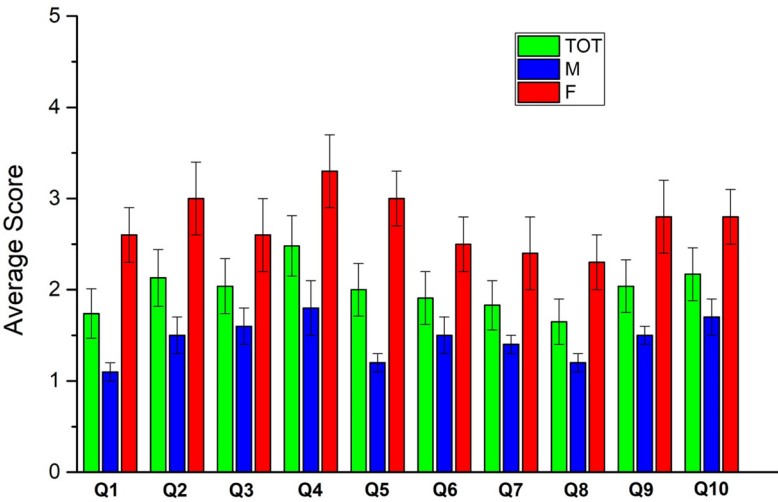

**Figure 4.** 2nd questionnaire: average score for each question (total, males, females).

### 3.3. Comparison of the Two Questionnaires

A significant improvement is observed for the whole set of patients for:

- total FAS score ($p < 0.001$), as well as mental ($p = 0.004$) and physical ($p < 0.001$) subsets;
- items 1 ($p < 0.001$), 2 ($p < 0.001$), 3 ($p = 0.007$), 4 ($p = 0.014$), 5 ($p = 0.009$), 8 ($p = 0.007$), 9 ($p = 0.004$)

No significant improvement was detected for items 6 ($p = 0.028$, $d = 0.50$, power = 0.49), 7 ($p = 0.134$, $d = 0.25$, power = 0.25) and 10 ($p = 0.113$, $d = 0.34$, power = 0.30)

All males reduced their FAS scores. The significant improvements are the following:

- total FAS score ($p < 0.001$), as well as mental ($p = 0.009$) and physical ($p < 0.001$) subsets;
- items 1 ($p < 0.001$), 2 ($p = 0.001$), 3 ($p = 0.012$), 4 ($p = 0.021$), 5 ($p = 0.018$), 8 ($p = 0.007$), 9 ($p = 0.017$)

Items 6 ($p = 0.119$, $d = 0.41$, power = 0.26), 7 ($p = 0.352$, $d = 0.21$, power = 0.13), 10 ($p = 0.184$, $d = 0.36$, power = 0.22) have no significant improvement. These items are related to mental fatigue.

For females the only significant improvement was found for item 3 ($p = 0.019$) of the questionnaire. No significant improvement was found for items 1 ($p = 0.032$, $d = 1.01$, power = 0.68), 2 ($p = 0.029$, $d = 1.15$, power = 0.78), 4 ($p = 0.154$, $d = 0.77$, power = 0.49), 5 ($p = 0.081$, $d = 0.77$, power = 0.49), 6 ($p = 0.056$, $d = 0.69$, power = 0.43), 7 ($p = 0.192$, $d = 0.34$, power = 0.17), 8 ($p = 0.106$, $d = 0.67$, power = 0.41), 9 ($p = 0.048$, $d = 0.97$, power = 0.65), 10 ($p = 0.224$, $d = 0.37$, power = 0.19). Non significant are also the improvements in FAS score ($p = 0.061$, $d = 0.87$, power = 0.57), and physical ($p = 0.052$, $d = 0.94$, power = 0.63) and mental ($p = 0.061$, $d = 0.76$, power = 0.48) fatigue. These results are related to the fact that 4 of the 10 females worsened their FAS scores while 6 reduced them. However, as the average variation in the FAS score for the 4 females that increased it is 4, while the average decrease for the remaining 6 is $-20.3$, it can be readily understood why most of the $p$ values are close to the significance threshold. As for those $p$ values the effect size is usually large ($>0.8$), an increase in the sample size in further studies can reduce the risk of a false negative conclusion.

The difference in the ability to reduce the FAS score between males and females (Table 4) is significant, despite the small size of the sample, as witnessed by the outcome of Fisher's Exact test: $p = 0.024$.

**Table 4.** Role of gender in the evolution of FAS score (2nd vs. 1st questionnaire). Number of patients is reported for each case.

|  | **M** | **F** | **Total** |
| --- | --- | --- | --- |
| increased | 0 | 4 | 4 |
| reduced | 13 | 6 | 19 |

### 3.4. Additional Outcomes

The smoking habitus (previous smokers vs no smokers) has no significant impact on the FAS in our group of patients. In fact for the 1st questionnaire $p = 0.440$, $d = 0.01$, power = 0.05 while for the 2nd questionnaire $p = 0.364$, $d = 0.01$, power = 0.08.

The Pearson's correlation coefficient between FAS score and BMI is $r(18) = 0.25$, $p = 0.288$ for the 1st questionnaire and $r(18) = 0.28$, $p = 0.23$ for the 2nd one. This shows that BMI has at most a marginal impact on the FAS score. The number of degree of freedom is limited to 18 because the BMI values of 3 patients were not available.

The Pearson's correlation coefficient between FAS score and age is $r(21) = -0.09$, $p = 0.683$ for the 1st questionnaire and $r(18) = 0.13$, $p = 0.55$ for the 2nd one. This shows that age has no impact on the FAS score in our case.

## 4. Discussion

Our outcomes show that gender is a significant parameter impacting the amount of fatigue and its evolution in time, while we cannot confidently rule out the place of care as another relevant parameter. Fatigue is present in most patients at the first screening and 4 of them, all females, worsen their score at the second screening, becoming candidates for a diagnosis of Chronic Fatigue. Although the knowledge of the mechanisms underlying fatigue is still limited [16] recent papers suggest relevant roles for

- the impaired mitochondrial function of neurons resulting from SARS-CoV-2 infection that compromise neurons with high metabolism energetic: a condition affecting cognitive process and inducing "brain fog", anergy and behavioral changes [17], and

- the proinflammatory cytokines, including interferon-gamma and interleukin-7, accumulated in the Central Nervous System, cross the blood-brain barrier in the circumventricular organs such as hypothalamus, leading to autonomic dysfunction with high fever as an acute symptom or long-lasting symptoms like alteration of the sleep-wake cycle, a cognitive dysfunction and a profound prolonged anergy [18].

Our results on the gender role highlight that females are impacted by Fatigue more significantly than males. In analyzing this outcome we need to take into account that males and females exhibit a different immunological response influenced by sex and gender. Males and females have a different level of risk of contracting COVID 19 acute Syndrome. The male gender is more susceptible to the development of severe acute illness [19] while the female one develops an autoantibody reaction unmasked by the virus that could play an important role in the genesis of Long COVID symptoms [20]. Moreover Chronic Fatigue Syndrome (CFS) has a higher incidence in the female gender as it happens for the post COVID 19 chronic fatigue [3]. This supports our findings on the role of gender.

Our outcomes are also in agreement with literature data that evidence that the positive clinical outcome rather than being the conclusive step of the disease can actually be the beginning of a lengthy and challenging path to full recovery [21]. In our study the CF prevalence is higher in females than males. The fact that home confined patients suffer from initially from a higher degree of fatigue is worth of further investigation with larger number of patients. Although possible reasons for this results will at this stage be purely speculative, it has already been reported that a relevant number of non hospitalized COVID-19 patients have not fully recovered their autonomy in everyday life [22]. Although both mental and physical fatigues are of impact in the lifestyle of patients, one interesting outcome of our study is that the physical fatigue is heavier than the mental one in the initial stages of recovery. Further studies are needed to investigate this point as to the best of our knowledge no report on this point was made. The full recovery of patients in fact needs personalized paths taking into account the details of the fatigue. Actions aimed to tackle physical fatigue (asthenia, muscular weakness) must be properly integrated with actions aimed to tackle the mental fatigue (anxiety, depression, feeling of abandonment) [23].

Finally, it is worth highlighting that 40% of the female patients but none of the males worsened their scores, becoming candidates for further investigation in the light of Long COVID syndrome.

## 5. Conclusions

We have reported about the prevalence and evolution of Fatigue on a small set of Italian patients that suffered from COVID-19 a few months after their clinical recovery. Our study, albeit carried on a limited number of patients, has shown that the prevalence of Fatigue is higher in females than in males, both for mental and physical fatigues. However, as most home confined patients were females and we cannot rule out the impact of place of care on fatigue, further work is needed to understand the specific roles of home confinement and gender. Our results highlight also the need to avoid the underestimation of the fatigue in home cared patients as they too require support to overcome the fatigue left behind by COVID-19. Lifestyle items, such as BMI and smoke habit have no significant impact in our study. Age also was not a relevant parameter.

As far as the evolution of recovery path is concerned it has to be highlighted that while all males improved their health, 4 over 10 females worsened their FAS score, which might indicate a development of Chronic Fatigue. Hence they are candidates for further assessments that might lead to a Long COVID and/or Chronic Fatigue diagnosis.

Studies involving larger number of patients are needed to verify on a larger scale and/or in different regions/countries the outcomes of our and other works. Moreover, multidisciplinary studies are needed as COVID-19 impacts on multiple organs, so that critical information about the medium and long term consequences of the COVID-19 syndrome can be understood and properly tackled.

**Supplementary Materials:** The following info are available online at https://www.mdpi.com/article/10.3390/covid1030044/s1, 1st questionnaire: Tables S1 (raw data); Table S2 (FAS score summaries); Table S3 (single items analysis); Table S4 (data from 2nd questionnaire); Table S5 (FAS score summary for the 2nd questionnaire); Table S6 (2nd questionnaire - score for each question (questions 4 and 10 are reverse scored)).

**Author Contributions:** All authors contributed to (i) conceptualization, (ii) brainstorming on outcomes (iii) manuscript writing and revisions. A.S., F.B., A.B., S.B., G.B., L.T., E.P., E.G., P.Z., C.D.M. collected the data during patients visits. A.T. carried out the statistical analysis. All authors have read and agreed to the published version of the manuscript.

**Funding:** This research received no external funding.

**Institutional Review Board Statement:** The study was conducted according to the guidelines of the Declaration of Helsinki, and approved by the CER (Comitato Etico Regionale—Regional Ethic Committee) of Liguria Region. Reference number of the study is 42/2021—DB id 11197.

**Informed Consent Statement:** Informed consent was obtained from all subjects involved in the study.

**Data Availability Statement:** All data from which the results were obtained are reported in Tables S1 and S2.

**Acknowledgments:** We wish to acknowledge the help given by medical office support staff.

**Conflicts of Interest:** The authors declare no conflict of interest.

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
