# Peer review of "Screening of a Small Number of Italian COVID-19 Syndrome Survivors by Means of the Fatigue Assessment Scale: Long COVID Prevalence and the Role of Gender"

_covid, doi:10.3390/covid1030044_

Round 1

Reviewer 1 Report

This is an interesting paper, with moderate clinical impact. It is well written, but presentation of results may be improved by replacing some of the tables by histograms. Table 2: "Ipothyroidism" should probably be "Hypothyroidism". What is the meaning of OSAS ?

Author Response

This is an interesting paper, with moderate clinical impact. It is well written, but presentation of results may be improved by replacing some of the tables by histograms. Table 2: "Ipothyroidism" should probably be "Hypothyroidism". What is the meaning of OSAS ?

We thank the referee for the appreciation of our work. We have replaced most tables with column graph and moved the tables to SI for completeness of information. We corrected the misspelling and explained the Acronym.

Reviewer 2 Report

In the article “Screening of Covid-19 syndrome survivors by means of Fatigue Assessment Scale: long Covid prevalence and the role of gender”, Serafini and colleagues examined the symptoms of mental and physical fatigue in a small sample of recovered COVID-19 patients in Italy. By comparing the participants’ responses to the Fatigue Assessment Scale questionnaire, the authors found that females tend to report higher fatigue than males in both the conducted screenings (one after the discharge and the other few months later). Also, home-confined COVID-19 patients tend to report higher fatigue when compared to the hospitalized ones in the first screening.

General Judgment Comments

Generally, the article is well written. The title clearly states the content of the paper, but it does not specify that the study is based on a small sample of the Italian population. Although the methods are appropriate, some details regarding the rationale for choosing the FAS questionnaire and the explanation of the Functional Limitation index are missing. Also, the statistical analysis needs to be adjusted for they lack rigor. For instance, when conducting the group comparisons, it is not clear whether the authors corrected the level of statistical significance or not. The power of tests is not reported as well, and it needs to be added, especially considering the small sample size of the study. Results should also be reported in the standard format.

For these reasons, I recommend the article to undergo Major Revision.

Major Issues

  • In the abstract, it is written that the paper aims to elucidate the role of age in the Long Covid fatigue, but this analysis is missing from the rest of the paper. Please provide information about it if age was tested in relation to the covid fatigue.
  • Why was the FAS questionnaire chosen over others in the literature to measure fatigue?
  • The FAS questionnaire is missing Cronbach’s alpha as an indicator of the test reliability. Please add it to the paper.
  • Given the small sample size of the study, knowing the power of the adopted statistical tests would be highly important. Please add it to the paper where needed.
  • From the text, it is not clear what the Functional Limitation Index is. Is it a standardized procedure or did the authors applied it ad-hoc to the items of the FAS questionnaire?
  • Lines 127, how did the authors test for normality of data distribution?
  • The paper focuses on the fatigue symptoms of a sample of COVID-19 survivors. A variable that could be interesting to explore would be the duration of COVID symptoms in the patients and its relationship with the fatigue symptoms in the subsequent period of time. Was this information available to the authors? If yes, could the authors provide more information about it, to see whether the groups were comparable?
  • The several multiple comparisons of scores between groups do not seem to be conducted by adjusting the level of statistical significance. Please adopt a statistical correction, such as Bonferroni’s to adjust the significance threshold.
  • Results are not reported in the standard format. Please also present the corresponding statistics and the p value for every test, for both significant and non-significant results.
  • Given the small sample size, please replace the Chi-Square tests with Fisher’s Exact tests.

Minor Issues

  • In the title, please highlight the fact that the article focuses on a small Italian sample of participants.
  • Introduction, line 56. Please change “In our country”, to “In Italy”.
  • Introduction, line 70. It is not necessary to report the p-value here.
  • Materials and Methods, line 85: please clarify how often was every follow-up visit conducted.
  • In Table 1, please clarify the full names of OSAS and COPD.
  • In Table 1, some percentage values are missing, please add them.
  • Please state which software was used to run the statistical analysis.
  • Materials and Methods lines 108-109: from the sentence “The FAS questionnaire […] is aimed to investigate the presence of fatigue in the Long Covid Syndrome […]” it seems that the FAS questionnaire was designed ad-hoc to measure the fatigue in the Long Covid Syndrome. Please rephrase the sentence.
  • Materials and Methods, line 112: please refer to ref. 7 in the format “Michielsen et al. (2003)”.
  • Materials and Methods, line 129: please clarify further comparisons or delete the “…” from the sentence “(previous smokers vs no smokers, male vs female, …).”
  • Results, lines 205-206: what is the p-value of this result? Is it significant or not?
  • Discussion, line 214: rather than “a number of them” report the exact number of patients with a worsened fatigue in the second screening.
  • Discussion, line 244: the authors stated that “physical fatigue was heavier than the mental one”. Did the authors test the difference between mental and physical fatigue under a statistical approach?
  • Conclusions, line 263: the authors stated that “Lifestyle items, such as BMI and smoke habit have no significant impact in our study”. Did the authors test the statistical significance of these results?

Final comments

I recommend the article undergo Major Revision. The manuscript tries to cover an important area of research but is missing some methodological details and needs further work, especially on the statistical analysis and the results format.

Round 2

Reviewer 2 Report

The article can now be accepted